# Inhibitory Effects of 2N1HIA (2-(3-(2-Fluoro-4-Methoxyphenyl)-6-Oxo-1(6H)-Pyridazinyl)-*N*-1H-Indol-5-Ylacetamide) on Osteoclast Differentiation via Suppressing Cathepsin K Expression

**DOI:** 10.3390/molecules23123139

**Published:** 2018-11-29

**Authors:** Sun-Hee Ahn, Zhihao Chen, Jinkyung Lee, Seok-Woo Lee, Sang Hyun Min, Nam Doo Kim, Tae-Hoon Lee

**Affiliations:** 1Department of Oral Biochemistry, Dental Science Research Institute, School of Dentistry, Chonnam National University, Gwangju 61186, Korea; sun3193@hotmail.com; 2Department of Molecular Medicine (BK21plus), Chonnam National University Graduate School, Gwangju 61186, Korea; chinaczhihao@gmail.com (Z.C.); wlsrud1945@naver.com (J.L.); 3Department of Dental Education and Periodontology, Dental Science Research Institute, School of Dentistry, Chonnam National University, Gwangju 61186, Korea; swlee@jnu.ac.kr; 4New Drug Development Center, Daegu-Gyeongbuk Medical Innovation Foundation, Dong-gu, Daegu 41061, Korea; shmin03@dgmif.re.kr; 5NDBio Therapeutics Inc., S24 Floor, Songdogwahak-ro 32, Yeonsu-gu, Incheon 21984, Korea; namdoo@ndbio.co.kr

**Keywords:** osteoclast, cathepsin K, osteoporosis, bone resorption

## Abstract

Osteoclasts are large multinucleated cells which are induced by the regulation of the receptor activator of nuclear factor kappa-Β ligand (RANKL), which is important in bone resorption. Excessive osteoclast differentiation can cause pathologic bone loss and destruction. Numerous studies have targeted molecules inhibiting RANKL signaling or bone resorption activity. In this study, 11 compounds from commercial libraries were examined for their effect on RANKL-induced osteoclast differentiation. Of these compounds, only 2-(3-(2-fluoro-4-methoxyphenyl)-6-oxo-1(6H)-pyridazinyl)-*N*-1H-indol-5-ylacetamide (2N1HIA) caused a significant decrease in multinucleated tartrate-resistant acid phosphatase (TRAP)-positive cell formation in a dose-dependent manner, without inducing cytotoxicity. The 2N1HIA compound neither affected the expression of osteoclast-specific gene markers such as TRAF6, NFATc1, RANK, OC-STAMP, and DC-STAMP, nor the RANKL signaling pathways, including p38, ERK, JNK, and NF-κB. However, 2N1HIA exhibited a significant impact on the expression levels of CD47 and cathepsin K, the early fusion marker and critical protease for bone resorption, respectively. The activity of matrix metalloprotease-9 (MMP-9) decreased due to 2N1HIA treatment. Accordingly, bone resorption activity and actin ring formation decreased in the presence of 2N1HIA. Taken together, 2N1HIA acts as an inhibitor of osteoclast differentiation by attenuating bone resorption activity and may serve as a potential candidate in preventing and/or treating osteoporosis, or other bone diseases associated with excessive bone resorption.

## 1. Introduction

Bone metabolism is an important process, as the continuous breaking down and building up involved in this procedure is essential for maintaining mineral homeostasis. The process of bone metabolism relies on the maintenance of a balance between the formation of bone matrix by osteoblasts and the elimination of mineralized bone by osteoclasts. The differentiation of osteoblasts and osteoclasts must, therefore, be critical for the homeostasis of bone.

Osteoclasts are the multinucleated cells (MNCs) responsible for bone resorption, a process whereby the hard, mineral component of bone, and the flexible soft-tissue component of bone are degraded. The process of osteoclast differentiation consists of the following steps: The differentiation of the osteoclast precursors into mononuclear preosteoclasts, fusion of the mononuclear preosteoclasts into multi-nuclear mature osteoclasts, and the activation of bone-resorbing osteoclasts. It is well known that the receptor activator of nuclear factor kappa-Β ligand (RANKL) and the macrophage colony-stimulating factor (M-CSF) in bone marrow-derived macrophage precursor cells are essential in the early stage of osteoclast differentiation [1,2]. In contrast, the late stage of differentiation in this lineage is characterized by the acquisition of mature phenotypic markers, such as the expression of tartrate-resistant acid phosphatase (TRAP), matrix metalloprotease-9 (MMP-9), and cathepsin K (CatK), as well as the morphological conversion to large multinucleated cells and the ability to form resorption lacunae on bones [3,4].

Osteoclasts are specialized cells for bone resorption and abnormal differentiation is highly correlated with various human bone diseases. The excessive activity of osteoclasts leads to osteoporosis, osteoarthritis, and rheumatoid arthritis, while osteoclast dysfunction leads to osteopetrosis [5,6,7]. Osteoporosis is characterized by low bone mass, structural deterioration, and porous bone, all of which are associated with higher fracture risk [8]. In recent years, osteoporosis has been recognized as an important public health issue in light of the increasing global life expectancy. An effective treatment for osteoporosis is therefore required. Molecular-level studies regarding the interaction of osteoclastic bone resorption within bone biology have led to the discovery of various therapeutic target compounds.

Several compounds have been reported to enhance bone formation by inhibiting the excess bone resorption capacity or promoting bone formation [9,10]. Compounds that influence the intracellular redox status can serve as regulators of bone formation and resorption by regulating redox-sensitive elements that are involved in the differentiation signaling pathway. Rotenone is one such compound, and it suppresses RANKL-induced osteoclast differentiation by regulating the mitogen-activated protein kinase (MAPK) signaling pathways [11]. Moreover, α-lipoic acid has been demonstrated to attenuate osteoclast differentiation by reducing NF-κB DNA binding and suppressing bone resorption in vivo [12]. Several small molecules or antibodies, such as the selective estrogen receptor modulators and RANKL antibodies mainly act by reducing osteoclast-mediated bone resorption. Bisphosphonates are the representative anti-resorption agents that bind to hydroxyapatite, reducing the number of osteoclasts and their activity [13]. They have been demonstrated to be effective in treating the progression of various bone loss-related diseases [14]. However, several side effects have been reported when bisphosphonates have been used.

In this study, we performed a biological screening using the in-house chemical library to identify small molecules for osteoclast target inhibition. We found that one chemical, 2-(3-(2-fluoro-4-methoxyphenyl)-6-oxo-1(6H)-pyridazinyl)-*N*-1H-indol-5-ylacetamide (2N1HIA), inhibited osteoclast differentiation and function. More importantly, this compound was demonstrated to suppress CatK expression and metalloprotease activity, both of which are involved in the late stage of osteoclast differentiation. Suppression of bone resorption and the pit-forming activity of osteoclast cells cultured on a calcium phosphate-coated plate were also observed when treated with 2N1HIA. Taken together, these results suggest that 2N1HIA suppresses bone resorption activity by regulating CatK expression.

## 2. Results

### 2.1. 2N1HIA Suppresses RANKL-Induced Osteoclastogenesis

As later described in the Materials and Methods section, we examined the effect of 11 compounds for their ability to cause osteoclast differentiation. To ensure that we selected the compounds having an inhibitory effect on osteoclastogenesis, we used bone marrow-derived macrophages (BMMs). The BMMs were incubated with each compound in the presence of RANKL and M-CSF, as indicated in Figure 1. Of the 11 chemicals, only chemical number one, 2-(3-(2-fluoro-4-methoxyphenyl)-6-oxo-1(6H)-pyridazinyl)-*N*-1H-indol-5-ylacetamide (2N1HIA) resulted in a significant decrease in the formation of multinucleated giant cells stained with TRAP. The other compounds, however, showed a similar level of differentiation (MNCs formation) to the positive control (dimethyl sulfoxide, DMSO) (Figure 1A,B).

To determine whether 2N1HIA suppresses osteoclastogenesis in a concentration-dependent manner, we first measured the formation and number of TRAP-positive MNCs in osteoclasts treated with 2N1HIA at the following concentrations: 0, 0.5, 1, 2, and 3 μM. As shown in Figure 2A, an evident reduction in multinucleated giant cells was observed when osteoclasts were treated with 2N1HIA; this decrease occurred in a concentration-dependent manner. Accordingly, the area of TRAP-positive MNCs and the number of TRAP-positive osteoclasts (cells with a nuclear number >3) were also significantly decreased when treated with 2N1HIA, in a concentration-dependent manner (Figure 2B,C). In the presence of 2N1HIA (1 μM), the number of MNCs induced by RANKL was reduced by 50%, whereas 3 μM of 2N1HIA completely abolished the formation of MNCs. In addition, the number of nuclei per osteoclast was dramatically reduced in a concentration-dependent manner when 2N1HIA was used as the treatment agent (Figure 2D). To exclude the possibility that the inhibition was due to cytotoxicity of 2N1HIA, cell viability was analyzed using a 3-(4,5-dimethylthiazol-2-yl)-2,5-diphenyltetrazolium bromide (MTT) colorimetric assay; 2N1HIA exhibited no cytotoxic effects after 72 h of treatment at concentrations of 0.5, 1, 2, and 3 μM (Appendix A). Moreover, 2N1HIA was not observed to affect M-CSF-mediated osteoclast precursor proliferation in the bromodeoxyuridine (BrdU) assay (Appendix A).

### 2.2. 2N1HIA Inhibits RANKL-Induced Cathepsin K Expression but Induces CD47 Expression

We examined the effect of 2N1HIA on the expression of various genes associated with osteoclast differentiation. The expression of the mRNA encoding osteoclast-specific markers such as NFATc1, OC-STAMP, DC-STAMP, and CatK markedly increased during RANKL-induced osteoclast differentiation (Figure 3). Although the addition of 2N1HIA did not affect the mRNA expression of TRAF6, NFATc1, RANK, OC-STAMP, and DC-STAMP (Figure 3A–E), it was observed to significantly suppress the RANKL-induced mRNA expression of CatK when compared to the control, DMSO (Figure 3F). This was highlighted on day three of differentiation, where the CatK expression after treatment with DMSO increased ~400-fold when compared to the expression level on day zero; CatK expression when treated with 1 μM of 2N1HIA however, increased ~250-fold when compared to its expression level on day zero. This result indicated that 1 μM of 2N1HIA treatment suppressed 40% of CatK expression when compared to the control that was treated with DMSO. A notable increase in CD47 expression (Figure 3H), an integrin-associated signal transducer membrane protein involved in cell adhesion to extracellular matrix [15,16], was observed when 2N1HIA was administered as the treatment agent. We also confirmed that the CatK protein (the proform and mature form) expression decreased in osteoclasts treated with 1 μM of 2N1HIA (Figure 4). As depicted in Figure 4A, CatK protein expression increased during RANKL-induced osteoclast differentiation. However, when treated with 2N1HIA, CatK protein expression significantly decreased; the same result was observed for the mRNA expression, as shown in Figure 3F. The CD47 protein expression was elevated by treatment with 2N1HIA compared to the control (DMSO treatment). Nevertheless, NFATc1 and TRAF6 protein expression was not affected by treatment with 2N1HIA; this result was consistent with those obtained from real-time PCR, as shown in Figure 3A. This was thought to occur because NFATc1 and TRAF6 are important osteoclastogenic makers. NFATc1 is known to be a master transcriptional regulator of osteoclast differentiation [17]. TRAF6 is known as a pivotal component in the RANK signaling pathway and an initiator in the activation of the NF-κB and MAPK signaling cascades [18].

To examine the effect of 2N1HIA on the signaling cascades downstream of RANK activity, phosphorylation of the signaling molecules in the MAPK and NF-κB pathways was detected. BMMs were pre-treated with 1 μM of 2N1HIA or control (DMSO) for 10 min and stimulated with RANKL at the indicated time points. As shown in Appendix A, the phosphorylation of the signaling molecules in the MAPK and NF-κB pathways was not altered in the presence of 2N1HIA, following RANKL stimulation.

To evaluate whether 2N1HIA treatment influences the activities of MMPs, which function as the representative proteases involved in bone resorption (along with CatK), we assessed the proteolytic activity of MMPs using zymography. Figure 4B shows that the activities of inactive and active MMP-9 were significantly reduced by treatment with 2N1HIA on days two and three of differentiation, respectively. These results suggest that 2N1HIA inhibits osteoclast differentiation by regulating CatK, CD47 expression, and MMP activity.

### 2.3. 2N1HIA Inhibits Bone Resorption Activity and Actin Ring Formation

The inhibitory effect of 2N1HIA on osteoclastic bone resorption activity was determined in the RANKL-induced osteoclasts that were cultured for five days, using a bone resorption assay kit (CosMo Bio, Tokyo, Japan). The plates were coated with fluoresceinamine-labeled chondroitin sulfate (FACS) and calcium phosphate (CaP). The mononuclear cells from the bone marrow were seeded in a complete α-MEM medium, supplemented with 100 ng/mL of mouse RANKL (mRANKL) and 25 ng/mL of recombinant mouse M-CSF for five days, with or without 3 μM of 2N1HIA. As the FACS bound to CaP was released from the CaP layer into the medium by osteoclastic resorption, the bone resorption activity was proportional to the fluorescence intensity of the FACS in the medium. As shown in Figure 5A, the differentiated osteoclasts with M-CSF and RANKL showed a strong fluorescence intensity of 6,000,000, whereas undifferentiated cells with only M-CSF did not exhibit a fluorescence intensity. Interestingly, 2N1HIA treatment dramatically suppressed the fluorescence intensity by 1,000,000. This indicated that 83% of bone resorption was inhibited by 3 μM of 2N1HIA treatment, relative to the control.

We also visualized the resorbed area on the plates under 100-fold magnification using a light microscope (Figure 5B). M-CSF and RANKL treatment markedly increased the number and size of bone resorption areas compared to that of M-CSF-only treatment (Figure 5B). Adding 2N1HIA however alleviated the number of bone resorption areas. To investigate the effect of 2N1HIA on actin ring formation, an immunofluorescence assay was performed (Figure 5C). Most of the RANKL-treated cells exhibited well-formed actin rings, whereas the cells treated with RANKL in the presence of 3 μM of 2N1HIA displayed no actin rings when compared to the cells that were only treated with RANKL.

### 2.4. 2N1HIA Has No Effect on Osteoblast Differentiation

It is known that some compounds with anti-osteoclastogenic effects also possess pro-osteoblastogenic activity [19]. To evaluate this possibility, we determined the effect of 2N1HIA on osteoblast differentiation. The results revealed that 2N1HIA did not affect alkaline phosphatase (ALP) staining activity or ALP mRNA expression induced by bone morphogenic protein (BMP) in primary osteoblasts (Appendix A). Thus, 2N1HIA appears to affect the differentiation of osteoclasts but not the differentiation of osteoblasts.

## 3. Discussion

Bones are constantly formed and reabsorbed by osteoblasts and osteoclast destruction, respectively [20]. These functions in bone regeneration are regulated by the bone microenvironment and the cellular signal transduction systems that regulate osteoblast and osteoclast differentiation [21]. Hyperactive osteoclasts are observed in many osteoclast-related diseases and are of a larger size than normal osteoclasts [22]. Osteoporosis is a serious disease commonly developed by postmenopausal women and older people in developed countries. In the elderly population, vertebral and hip fractures are chronic conditions in which the activity of osteoclasts and osteocytes is abnormally balanced; thus, the balance of bone destruction and bone resorption plays a very important role in these conditions and is considered a therapeutic goal in these patients [23,24]. Many reports have suggested that several compounds ameliorate bone loss in osteoporosis by inhibiting the signaling pathways for osteoclast differentiation [25,26,27]. In this study, we examined the inhibitory effects of 2N1HIA on in vitro RANKL-induced osteoclast differentiation. This study is the first to present that 2N1HIA inhibits RANKL-stimulated osteoclastogenesis by suppressing the expression of the late differentiation marker, CatK, and the RANKL-induced osteoclast function by disturbing actin ring formation and decreasing MMP activity.

MAPK-NFATc1 constitutes the main downstream signaling cascade of RANKL in the early stage of osteoclast differentiation, whereas TRAP and CatK are required for bone resorption during late-stage osteoclastogenesis. It is well known that RANKL signaling rapidly triggers MAPKs such as p38, extracellular signal-regulated kinase (ERK), and c-Jun *N*-terminal kinase (JNK) during osteoclastogenesis [28,29]. In this study, when 2N1HIA was added to the cultures, these signaling pathways were not influenced, and neither were the osteoclasts or NF-kB signaling (Appendix A). Moreover, the expressions of other genes related to the early stage of osteoclast differentiation such as TRAP6, NFATc1, RANK, OC-STAMP, and DC-STAMP were not altered when treated with 2N1HIA. Interestingly, only CatK and CD47 gene expressions were significantly affected by 2N1HIA treatment (Figure 3), an observation supported by the protein expression results (Figure 4A).

CD47, a member of the immunoglobulins superfamily, is known as a fusion factor involved in the multinucleation process in osteoclasts [30]. Hobolt-Pederson et al. found that cells presenting CD47 appeared relatively small compared to CD47-negative osteoclasts. They also observed that CD47-positive osteoclasts had significantly fewer nuclei than the CD47-negative osteoclasts, and adding a CD47-blocking antibody resulted in an increased formation of large nuclei-rich osteoclasts. Taken together, their results indicate that the CD47-presenting osteoclasts are smaller than CD47-negative cells based on cell size and number of nuclei. In our study, 2N1HIA treatment increased CD47 expression in osteoclasts and inhibited osteoclast differentiation. More precisely, treatment with 2N1HIA suppressed the resorption activity, which is a responsibility of CatK.

Hobolt-Pederson et al. also reported that in the presence of CD47, a negative correlation exists with the osteoclast marker CatK in the late stage of differentiation. This is because the result more inclined to being observed was a lower proportion of cells presenting CD47 than the proportion of osteoclasts expressing CatK; a significant correlation was also observed. In other words, a higher proportion of cells presenting CD47 reduced the proportion of osteoclasts expressing CatK. Interestingly, our findings of higher CD47 proportion with a lower CatK expression (Figure 4A) align with their result. The precise mechanism by which 2N1HIA regulates the expression of CD47 and CatK remains unclear.

Osteoclasts have the unique property of bone resorption; therefore requiring proteinases to efficiently degrade the bone matrix [31]. The activities of osteoclasts, including migration, anchoring to the bone surface, and bone resorption are well known processes in which proteinases play an important role [32]. Bone resorption is determined by the proteolytic activities that occur well before bone matrix degradation is initiated [33]. The most prominent proteinases for osteoclast function belong to the cysteine proteinase and MMP groups [34]. The critical cysteine proteinase is CatK, which limits the solubilization of osteoclast bone matrices [35]. The expression of CatK is also high in osteoclasts when compared to other cells [36]. In this study, we observed an increased expression of CatK during osteoclast differentiation (Figure 3F and Figure 4A) and the inhibited expression of CatK, the lysosomal enzyme responsible for bone resorption, due to the addition of 2N1HIA. Inhibiting CatK expression could serve as an effective way to reduce bone resorption activity. This is as CatK is relevant in pathological situations of bone resorption such as osteoporosis, and the selective inhibitors of CatK inhibit bone resorption in animal models of osteoporosis [37].

Osteoclasts express a variety of proteases including cathepsins and MMPs [31]. However, it is generally known that CatK is the major bone-degrading enzyme [38]. According to Schurigt et al. [39], CatK protein cleaves and activates MMP-9 in acidic environments as observed in bone resorption. These authors therefore proposed that a key link exists between CatK expression in bone and the extracellular cell matrix remodeling through MMP-9 activation. This mechanism may be used to explain our observation regarding the inhibition of CatK expression and the attenuation of MMP activity.

MMPs are potential contributors to bone destruction [39], and are critical to the access of future resorption sites by osteoclasts. This was demonstrated in a study showing that the MMP inhibitors completely prevented the formation of the marrow cavity of primitive long bones [31]. When there is a deficiency in MMP-9, the migration of osteoclasts is dramatically slowed down [40]. However, this effect pertains to migration and not resorption, which was made clear as the strong inhibitors of bone resorption did not prevent the recruitment of cells in the diaphysis [31]. In this study, we found that 2N1HIA significantly inhibited MMP-9 activity and decreased the area of bone lacunae. Based on previous reports pertaining to the slowing down in migration of osteoclasts under MMP-9-deficient conditions, we expected that a decreased level of MMP-9 in osteoclasts due to 2N1HIA treatment would not be restricted to bone resorption, but would also affect the migration of osteoclasts. Therefore, extending MMP-9’s effect to migration may then contribute to osteoclast recruitment and in vivo bone loss (anti-osteoporosis effect).

The study conducted had a few limitations. Although it is known that treating osteoclasts with 2N1HIA inhibits their differentiation by increasing CD47 expression and decreasing CatK expression, the exact regulation mechanism of 2N1HIA on the expression of these genes in osteoclasts remains unclear. In addition, we only measured the effect of 2N1HIA on osteoclast differentiation and bone resorption in vitro, which makes our study limiting. Therefore, identifying whether administering 2N1HIA at a clinical dosage can exert an anti-osteoporosis effect in animals and humans requires further investigation.

To conclude, based on our study findings, 2N1HIA attenuated RANKL-induced osteoclast differentiation and bone resorption via suppressing CatK and MMP expression, without exhibiting any cytotoxic effect. The compound, 2N1HIA, could therefore serve as a promising candidate in the development of a therapeutic approach to treating bone resorption diseases caused by RANKL. Further studies to test and develop this therapeutic approach for resorption diseases are also warranted.

## 4. Materials and Methods

### 4.1. Ethics Statements

Mice were housed in a specific pathogen-free facility, following the guidelines provided in the Guide for the Care and Use of Laboratory Animals (Chonnam National University, Gwangju, Korea). Adult male C57BL/6J mice (8 weeks old) were used in this study. The collection of primary mononuclear cells from mice was approved by the IACUC at Chonnam National University (Approval No. CNU IACUC-YB-2017-70, 31 October 2017).

### 4.2. Chemical Library

We performed a biological screening with our in-house synthetic compounds. Our chemical library source consisted of commercially available compounds (Hit2lead, San Diego, CA, USA). Detailed information regarding the compounds used in this study is provided in Appendix A. These compounds were dissolved in DMSO, and aliquots of the compound solutions were stored at −20 °C and diluted to the appropriate concentration before use.

### 4.3. Cell Viability Assay

To evaluate the cytotoxicity of 2N1HIA, we used the MTT colorimetric assay to measure cell viability following treatment with 2N1HIA, as described previously [41,42]. Briefly, osteoclasts were cultured in a 96-well plate, with or without 2N1HIA for 3 days, the MTT labeling reagent (0.5 mg/mL final concentration) was then added to each well. The cells were incubated for 4 h, followed by the addition of 100 μL of 10% SDS in 0.01 M of hydrochloric acid solution, and further incubated for another 3 h. The absorbance at 450 nm was measured using a microplate reader (Molecular Devices, San Jose, CA, USA, model: SpectraMax i3x). The reported values represent the means of the triplicates and are expressed as a percentage of the control values: (A_450_ of 2N1HIA treatment/A_450_ of control) × 100.

### 4.4. Osteoclast Precursor Proliferation Assay

Osteoclast precursor proliferation was quantified using a BrdU Cell Proliferation Assay kit (GE Healthcare Life Sciences, Piscataway, NJ, USA) [43]. BMMs were treated with 30 ng/mL of M-CSF (PeproTech, Rocky Hill, NJ, USA) for 3 days in the presence or absence of 2N1HIA. Cell proliferation was quantified from BrdU incorporation using the BrdU ELISA assay kit (Cell Signaling Technology, Boston, MA, USA).

### 4.5. In Vitro Osteoclastogenesis Assay

For the in vitro osteoclast differentiation, mouse bone marrow cells were isolated from the femurs and tibiae of 8-week-old mice, by flushing the bone marrow with α-minimum essential medium (MEM). The flushed cells were incubated in a culture medium (α-MEM containing 10% heat-inactivated fetal bovine serum (FBS) and 1% penicillin/streptomycin) for 1 day. The suspended cells were then collected and cultured in a medium supplemented with 30 ng/mL of M-CSF (PeproTech, Rocky Hill, NJ, USA) for 3 days in order to obtain BMMs. Thereafter, non-adherent cells were discarded and the adherent cells further cultured in the medium were supplemented with 30 ng/mL of M-CSF and 50 ng/mL of RANKL (PeproTech, Rocky Hill, NJ, USA) for up to 5 days. The medium was replenished every 2 days. To assess the extent of differentiation, the differentiated BMMs were fixed for 30 min in 3.7% formaldehyde in phosphate-buffered saline (PBS), and stained using a TRAP kit (Sigma-Aldrich, St. Louis, MO, USA). TRAP-positive multinucleate cells containing five or more nuclei were counted. After four days, the mature osteoclasts were enumerated under a microscope based on the number of nuclei (n ≥ 3), cell size and cell number. Each osteoclast formation assay was performed independently at least 3 times.

### 4.6. In Vitro Osteoblastogenesis Assay

Primary mouse osteoblasts were isolated from the calvaria of 3-day-old C57BL/6J mice by sequential digestion with collagenase, as previously described [44]. The primary osteoblasts were cultured in α-MEM containing 10% heat-inactivated FBS and 1% penicillin/streptomycin. After 3 days, the cells were reseeded (4 × 10^3^ cells/well) and cultured in an osteogenic medium containing 100 ng/mL of bone morphogenic protein 2 (BMP-2) (Sino Biological, Wayne, PA, USA) with different concentrations of 2N1HIA (0.5, 1, 2, and 3 μM) or without it. The medium was exchanged for a fresh medium every 2 days. After 1, 7, or 14 days, alkaline phosphatase (ALP) staining was performed. For ALP staining, cells were fixed with 4% formalin for 10 min and stained for 30 min with an ALP staining solution, in accordance with the manufacturer’s instructions (Sigma-Aldrich, St. Louis, MO, USA). Each osteoblast differentiation assay was performed independently at least three times.

### 4.7. Real-Time PCR

Using the QIAzol RNA Lysis reagent (Qiagen Sciences, Valencia, CA, USA), total RNA was isolated from BMMs treated with M-CSF and RANKL or primary osteoblasts treated with BMP-2. The complementary DNAs were then synthesized using a PrimeScript™ RT Reagent Kit for the real-time PCR assay (Takara Biotechnology, Tokyo, Japan), according to the manufacturer’s instructions. Quantitative PCR was performed using a QuantStudio 3 real-time PCR system (Applied Biosystems, Foster City, CA, USA) with a Power SYBR Green PCR Master Mix (Applied Biosystems, Foster City, CA, USA), and a standard temperature protocol. The results obtained using a cycle threshold are expressed as relative quantities and were calculated using the 2^−ΔΔ^CT method (expressed as the relative fold ratio). Glyceraldehyde 3-phosphate dehydrogenase (GAPDH) served as the control gene for normalization. Three separate experiments were performed. Appendix A lists the primers used for the quantitative real-time PCR assay.

### 4.8. Western Blotting

Osteoclast cells were lysed in a chilled lysis buffer (50 mM Tris–HCl (pH 7.5), 150 mM NaCl, 1% NP-40, 0.5% sodium deoxycholate, 0.1% sodium dodecyl sulfate (SDS), 2 mM ethylenediaminetetraacetic acid (EDTA), and protease inhibitors), and the supernatants were collected following centrifugation (10,000× *g*, 4 °C, 30 min). The concentration of protein extracted from the differentiated osteoclasts was determined using BCA protein assays (Pierce, Rockford, IL, USA). Protein samples (30 μg) were subjected to a 12% polyacrylamide gel electrophoresis and transferred to a polyvinylidene fluoride membrane. The membranes were then incubated with anti-β-actin (Sigma-Aldrich, St Louis, MO, USA), anti-cathepsin K (Santa Cruz Biotechnology, Dallas, TX, USA), anti-CD47 (Abcam, Cambridge, MA, USA), anti-TRAF6 (Santa Cruz Biotechnology, Dallas, TX, USA), anti-NFATc1 (Cell Signaling Technology, Boston, MA, USA), anti-ERK1/2 (Cell Signaling Technology, Boston, MA, USA), anti-phospho-ERK1/2 (Cell Signaling Technology, Boston, MA, USA), anti-p38 (Cell Signaling Technology, Boston, MA, USA), anti-phospho-p38 (Cell Signaling Technology, Boston, MA, USA), anti-JNK (Cell Signaling Technology, Boston, MA, USA), anti-phospho-JNK (Cell Signaling Technology, Boston, MA, USA), anti-p65 (Cell Signaling Technology, Boston, MA, USA), or anti-phospho-p65 (Cell Signaling Technology, Boston, MA, USA). This was followed by incubation with the horseradish peroxidase (HRP)-conjugated secondary antibody (Cell Signaling Technology Boston, MA, USA) which was detected using an ECL system (iNtRON, Seoul, Korea).

### 4.9. Resorption Pit Assay

For the bone resorption activity assay, a bone resorption assay kit (CosMo Bio, Tokyo, Japan) was used, according to the manufacturer’s directions. BMMs were cultured on a bone resorption assay plate 48 (2 × 10^4^ cells/well), with or without 1 μM of 2N1HIA, in the presence of M-CSF (30 ng/mL) and RANKL (50 ng/mL) for 6 days. On the seventh day following cell seeding, the culture supernatant was harvested into a 96-well black polypropylene micro-well plate (Thermo Fisher Scientific Nunc, Waltham, MD, USA) and mixed with 50 μL of 0.1 N NaOH. The fluorescence intensity was then measured using a fluorescence plate reader (Molecular Devices, San Jose, CA, USA, model: SpectraMax i3x); the excitation and emission wavelengths were 485 nm and 535 nm, respectively. In addition, the pit areas were calculated by considering ten randomly selected pictures per well, taken at 10 times magnification with the ImageJ software (https://imagej.nih.gov/ij). The total resorption area was measured for each of the random pictures and the mean resorption area was posteriorly calculated.

### 4.10. Zymography

The activity of proteolytic enzymes was evaluated using gelatin, following the electrophoretic separation of lysates from the BMMs culture, with or without 1 μM of 2N1HIA, in the presence of M-CSF (30 ng/mL) and RANKL (50 ng/mL) for 5 days. Briefly, cells (6 well plate) were mixed with a homogenization buffer (50 mM Tris-HCl, pH 6.8, 150 mM NaCl, and 1% Triton X-100) and the lysates were collected following centrifugation (5000× *g*, 10 min, 4 °C). Thirty micrograms of unheated and non-denatured protein were subjected to 0.1% gelatin (Sigma) containing SDS-polyacrylamide gel electrophoresis. Gels were washed in a renaturing buffer (2.5% Triton X-100) and incubated in a Novex zymogram developing buffer (Invitrogen, Carlsbad, CA, USA) for 16 h at 37 °C. Gels were stained with 0.2% Coomassie brilliant blue R (Sigma, St. Louis, MO, USA) for 1 h, and de-stained in 20% methanol and 10% glacial acetic acid.

### 4.11. Actin Ring Formation Assay

Actin rings of osteoclasts were detected by staining the actin filaments with rhodamine-conjugated phalloidin. Osteoclasts were formed from the BMMs culture in the presence of M-CSF (30 ng/mL) and RANKL (50 ng/mL), with or without 2N1HIA (3 μM). At the end of the incubation, osteoclasts were stained with rhodamine-conjugated phalloidin to observe the actin. The distribution of actin rings was visualized and detected using a fluorescence microscope.

### 4.12. Statistical Analysis

Statistical analyses were performed using unpaired two-tailed Student’s *t* tests (* *p* < 0.05; ** *p* < 0.01; *** *p* < 0.001; NS, not significant). All data are expressed as the mean ± standard deviation (SD). Results are representative examples of more than three independent experiments.

## Figures and Tables

**Figure 1 molecules-23-03139-f001:**
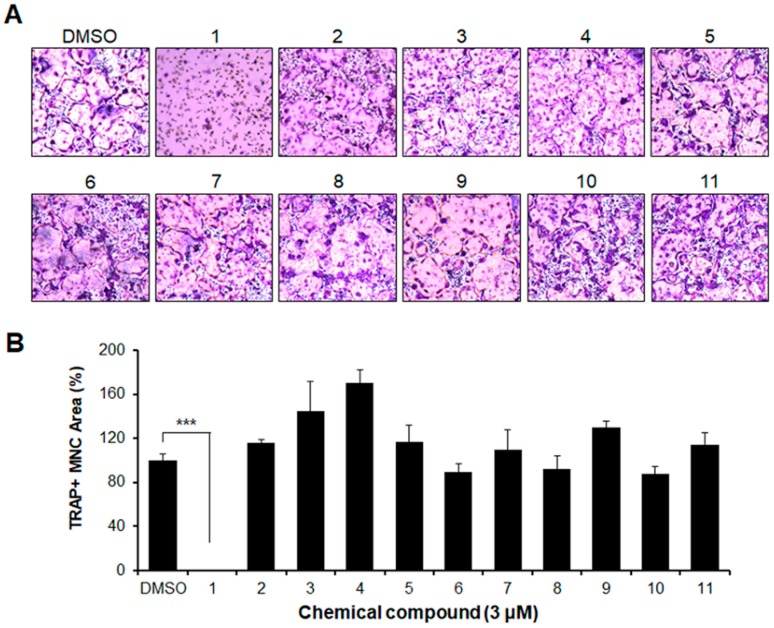
Selection of the inhibitor for osteoclast differentiation from eleven compounds (1 to 11) as shown in Appendix A. Bone marrow-derived macrophages were cultured with a macrophage colony-stimulating factor (30 ng/mL) and the receptor activator of nuclear factor kappa-Β ligand (50 ng/mL) in the presence of each chemical compound or the control, dimethyl sulfoxide. After four days, the cells were fixed with 3.7% formalin, permeabilized with 0.1% Triton X-100, and stained for tartrate-resistant acid phosphatase (TRAP). TRAP-positive cells (nuclear number >3) were counted as osteoclasts. (**A**) TRAP staining patterns of osteoclasts are shown. (**B**) The TRAP and multinucleated cells were counted based on the number of nuclei in each cell; the percentage of cells with the indicated range of nuclei per cell was calculated. *** *p* < 0.001 indicate statistical significance.

**Figure 2 molecules-23-03139-f002:**
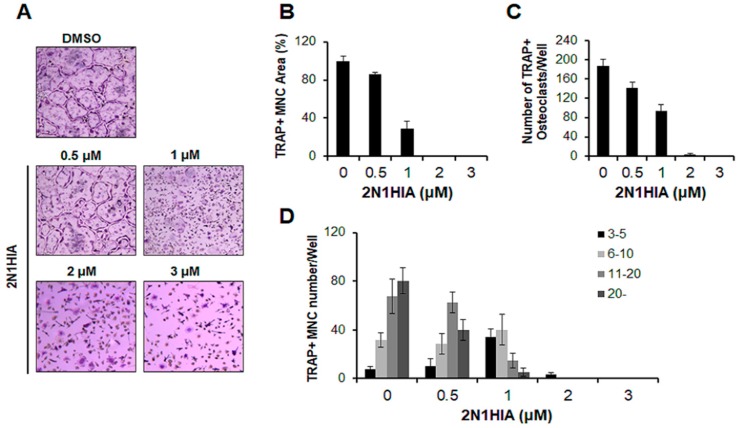
Effect of 2-(3-(2-fluoro-4-methoxyphenyl)-6-oxo-1(6H)-pyridazinyl)-*N*-1H-indol-5-ylacetamide (2N1HIA) on the receptor activator of nuclear factor kappa-Β ligand (RANKL)-induced osteoclast differentiation. Bone marrow-derived macrophages were cultured with a macrophage colony-stimulating factor (30 ng/mL) and RANKL (50 ng/mL) in the presence of 2N1HIA or the control, dimethyl sulfoxide (DMSO). Cells were stained for the tartrate-resistant acid phosphatase (TRAP) activity, and both TRAP and multinucleated cells (MNCs) were counted. (**A**) The TRAP staining patterns of osteoclasts are shown. (**B**) The area of TRAP-positive cells (nuclear number >3) was calculated as a relative value by comparing it to that of DMSO (100%). (**C**) TRAP-positive cells with a nuclear number >3 were counted as osteoclasts. (**D**) The TRAP and MNCs were counted based on the number of nuclei in each cell; the percentage of cells with the indicated range of nuclei per cells was calculated. The data are representative of three independent experiments and presented as the mean ± standard deviation.

**Figure 3 molecules-23-03139-f003:**
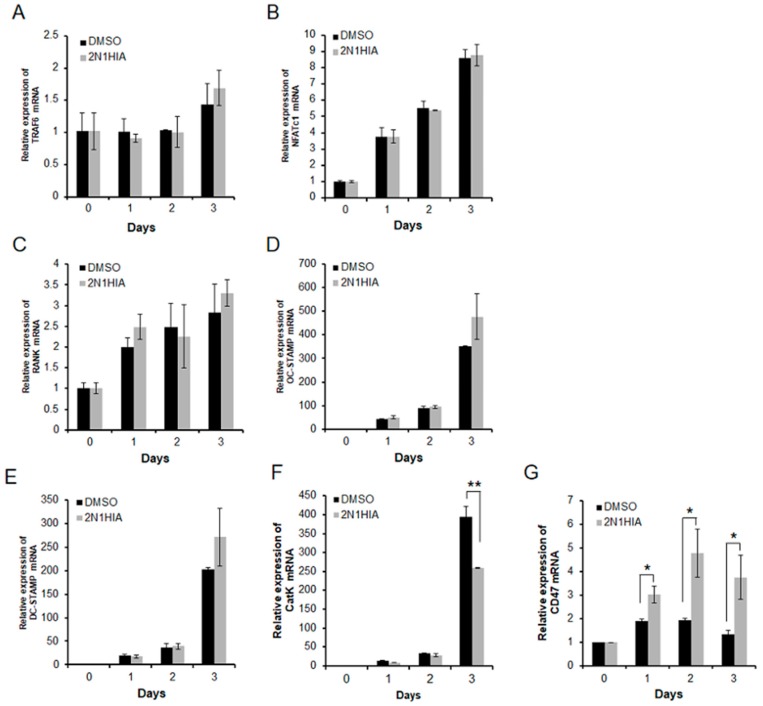
Effect of 2N1HIA on the expression of osteoclast-related genes. Bone marrow-derived macrophages were cultured in the presence of the receptor activator of nuclear factor kappa-Β ligand (RANKL) and the macrophage colony-stimulating factor together, with or without 1 μM of 2N1HIA. After cultivation for the indicated time periods, the expression of (**A**) TRAF6, (**B**) NFATc1, (**C**) RANK, (**D**) OC-STAMP, (**E**) DC-STAMP, (**F**) CatK, and (**G**) CD47 mRNA was analyzed by real-time PCR. The expression levels were normalized to GAPDH and expressed relative to day zero. Statistical analysis was performed between the control (dimethyl sulfoxide) and the 2N1HIA-treated values at the indicated time points; * *p* < 0.5; ** *p* < 0.01 indicate statistical significance.

**Figure 4 molecules-23-03139-f004:**
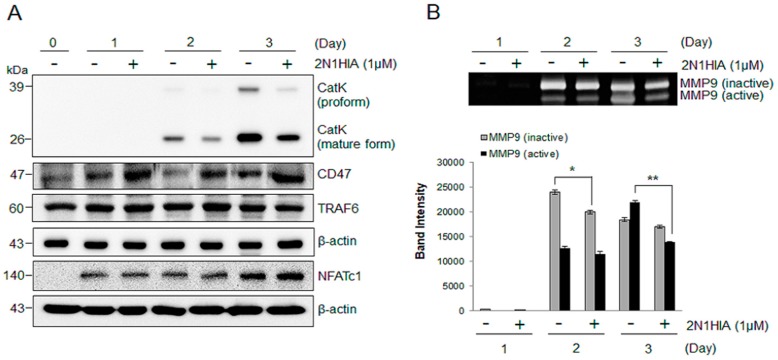
Effect of 2N1HIA on the protein expression of osteoclast-related targets. Bone marrow-derived macrophages were cultured in the presence of both the receptor activator of nuclear factor kappa-Β ligand and the macrophage colony-stimulating factor, with or without 1 μM of 2N1HIA. After cultivation for the indicated time periods, the protein expression of CatK, CD47, and TRAF6 was analyzed by Western blot (**A**). Matrix metalloprotease-9 activity was assessed by zymography and densitometric analysis was performed by the Image J software (**B**); * *p* < 0.5; ** *p* < 0.01 indicate statistical significance.

**Figure 5 molecules-23-03139-f005:**
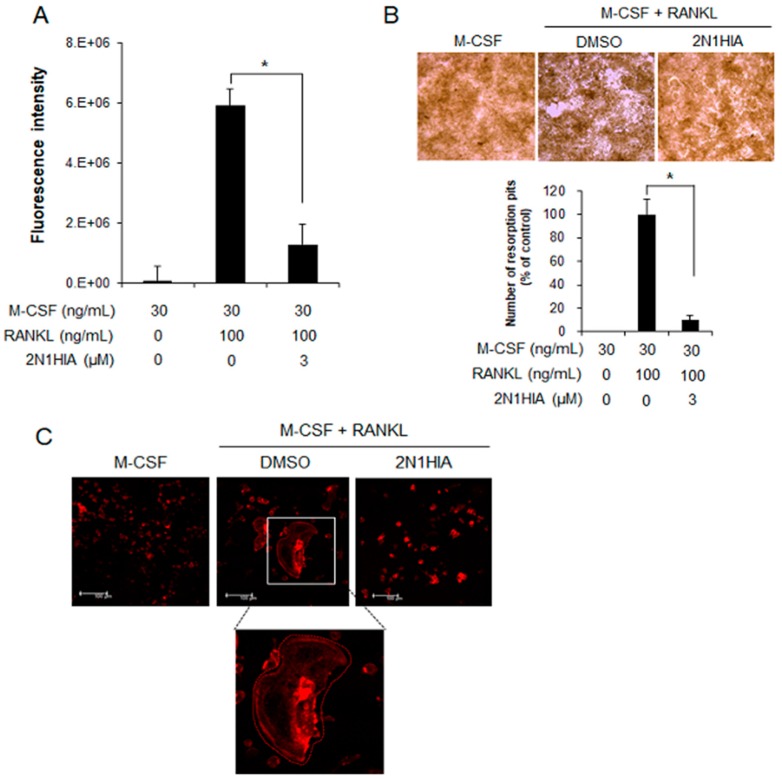
Resorption activity of osteoclasts on bovine bone slices, with or without 2N1HIA. (**A**) The resorption index of osteoclasts differentiated from bone marrow with dimethyl sulfoxide (DMSO) (control) or treated with the 2N1HIA. Fluorescence intensity was measured at an excitation wavelength of 485 nm and an emission wavelength of 535 nm using a fluorometric plate reader. Data are expressed as percentages of the values of untreated cells (mean ± standard deviation, *n* = 3). (**B**) The visualized absorbed area with 2N1HIA treatment. Quantitative analysis of the pit area of the osteoclasts on calcium phosphate-coated plates, with and without 3 µM of 2N1HIA. The percentage area covered by resorption pits relative to untreated control was measured using the ImageJ software (version 1.41). A test was performed to evaluate significant differences. Three distinct sample pools were analyzed in a duplicate manner (*n* = 6). * *p* value < 0.05 is considered statistically significant compared the control. (**C**) Visualization of actin ring formation in RANKL-induced osteoclasts in the presence of 3 µM of 2N1HIA or the control, DMSO. The receptor activator of nuclear factor kappa-Β ligand-induced osteoclasts were cultured in the presence of 3 µM of 2N1HIA for four days. Cells were fixed and stained for actin rings as described in the Materials and Methods section. The data shown are representative of at least two independent experiments.

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
