# Peer review of "Inhibitory Effects of 2N1HIA (2-(3-(2-Fluoro-4-Methoxyphenyl)-6-Oxo-1(6H)-Pyridazinyl)-N-1H-Indol-5-Ylacetamide) on Osteoclast Differentiation via Suppressing Cathepsin K Expression"

_molecules, 2018, doi:10.3390/molecules23123139_

Round 1
Reviewer 1 Report
In this paper, the authors showed the inhibitory effects of 2N1HIA on osteoclast differentiation. The 2N1HIA triggered the elevation in CD47 expression and down regulation of cathepsinK, simultaneously. Although the novelty as well as significance of this study is high, several issuses should be clarified before publication in Molecules.
1. The authors determined the cytotoxicity of 2N1HIA by MTT assay. However, the MTT assay could not exactly reflect the genuine effects of 2N1HIA on cell proliferation. Thus the authors should examine whether 2N1HIA affects precursor cell prolifaration by BrdU method.
2. In this paper, the major pathway results were obtained by PCR analysis. As the authors have addressed, one of the most important osteoclastogenic transcription factor is NFATc1, which has a distinctive intracellular translocation at the downsteam of RANKL stimulus. It is highly recommended that the authors should examine the protein expression level of NFATc1 by W.B. in the revised manuscript.
Author Response
Dear Reviewer #1,
We thank you for your careful review of our manuscript and for providing us with comments and suggestions to improve the quality of the manuscript.
Responses to reviewer #1 specific comments
1. The authors determined the cytotoxicity of 2N1HIA by MTT assay. However, the MTT assay could not exactly reflect the genuine effects of 2N1HIA on cell proliferation. Thus the authors should examine whether 2N1HIA affects precursor cell prolifaration by BrdU method.
=> We appreciate your comment. In response, we have examined the effect of 2N1HIA on precursor cell proliferation by using the BrdU method (Supplementary Figure S1B). The results suggest that the inhibitory effect of 2N1HIA on osteoclasts was not due to the proliferation effect. We have included these data and revised the manuscript accordingly.
Page 5, lines 147-149; Page 11, lines 436-440, Supplemental Figure S1B
2. In this paper, the major pathway results were obtained by PCR analysis. As the authors have addressed, one of the most important osteoclastogenic transcription factor is NFATc1, which has a distinctive intracellular translocation at the downsteam of RANKL stimulus. It is highly recommended that the authors should examine the protein expression level of NFATc1 by W.B. in the revised manuscript.
=> We appreciate your comment and agree with this point. In response, we have performed a western blotting analysis for NFATc1. We have included these data in Figure 4A and revised the manuscript accordingly.
Page 6, lines 183-188, Figure 4A
We hope our revised manuscript is now suitable for publication in Molecules and that it will be helpful to other investigators who are conducting similar research on subjects such as osteoclast differentiation related to bone diseases.
This manuscript has not been published or presented elsewhere in part or in entirety and is not under consideration by another journal. The study design was approved by the appropriate ethics review board. We have read and understood your journal’s policies, and we believe that neither the manuscript nor the study violates any of these. The authors declare no conflicts of interest.
Thank you for your consideration of our manuscript; I look forward to hearing from you.
Sincerely,
Tae-Hoon Lee, Ph.D.
Reviewer 2 Report
The current paper is technically sound providing convincing evidences for the proof of concept.
Further, the current work carried out in the manuscript is of clinical relevance. However, there are certain concerns that needs to be worked out.
There is a coupling role between the osteoclasts and the osteoblasts. The authors have not performed any such experiments in the current manuscript.
Author Response
Dear Reviewer #2,
We thank you for your careful review of our manuscript and for providing us with your comments and suggestions to improve the quality of the manuscript.
Responses to reviewer #2 specific comments
There is a coupling role between the osteoclasts and the osteoblasts. The authors have not performed any such experiments in the current manuscript.
=> Good point. We agree with your point. Actually, we did perform an osteoblastogenesis assay but did not find any impact of 2N1HIA on osteoblast differentiation. We did not include these data in our previous manuscript. In response to the reviewer’s comment, we have included these data in Supplementary Figure 3S and revised the manuscript accordingly.
Page 9, lines 314-320, Page 12, lines 458-468, Supplemental Figure S3
We hope our revised manuscript is now suitable for publication in Molecules and that it will be helpful to other investigators who are conducting similar research on subjects such as osteoclast differentiation related to bone diseases.
This manuscript has not been published or presented elsewhere in part or in entirety and is not under consideration by another journal. The study design was approved by the appropriate ethics review board. We have read and understood your journal’s policies, and we believe that neither the manuscript nor the study violates any of these. The authors declare no conflicts of interest.
Thank you for your consideration of our manuscript; I look forward to hearing from you.
Sincerely,
Tae-Hoon Lee, Ph.D.
Round 2
Reviewer 1 Report
The manuscript has been improved to be published in Moleculs.
Reviewer 2 Report
The authors have performed the necessary amendments in the revised version. Overall, the manuscript is well presented in the current form.